# Compressible Cellulose Wood Prepared with Deep Eutectic Solvents and Its Improved Technology

**DOI:** 10.3390/polym15071593

**Published:** 2023-03-23

**Authors:** Wenhao Wang, Mengyao Chen, Yan Wu

**Affiliations:** College of Furnishings and Industrial Design, Nanjing Forestry University, Nanjing 210037, China

**Keywords:** wood chemistry, deep eutectic solvents, cellulose nanofiber, mechanical compressibility, elastic porous structure

## Abstract

Elastic materials have a wide range of applications in many industries, but their widespread use is often limited by small-scale production methods and the use of highly polluting chemical reagents. In this study, we drew inspiration from research on wood softening to develop an environmentally friendly and scalable approach for producing a new type of compressible wood material called CW from natural wood. To achieve this, we employed a top-down approach using a novel type of “ionic liquid” eutectic solvent (DES) that is cost-effective, environmentally friendly, and recyclable. After treatment with DES, the resulting CW demonstrated good elasticity and durable compressibility, which was achieved by removing some lignin and hemicellulose from the wood and thinning the cell walls, thereby creating a honeycomb structure that allows for sustained compression and rebound. However, we found that the wood treated with a single eutectic solvent showed some softening (CW-1), although there was still room for further improvement of its elasticity. To address this, we used a secondary treatment with sodium hydroxide alkali solution to produce a softer and more elastic wood (CW-2). We conducted a series of comparative analyses and performance tests on natural wood (NW) and CW, including microscopic imaging; determination of chemical composition, mechanical properties, and compressive stress effects; and laser confocal testing. The results show that the DES and sodium hydroxide alkali solution treatments effectively removed some lignin, hemicellulose, and cellulose from the wood, resulting in the thinning of the cell walls and creating a more elastic material with a sustainable compression rebound rate of over 90%. The various properties of CW, including its elasticity, durability, and sustainability, provide great potential for its application in a range of fields, such as sensors, water purification, and directional tissue engineering.

## 1. Introduction

Cellular materials, especially those that are hydrated, have garnered significant research interest due to their unique characteristics. These materials have the ability to contain water, are biocompatible and environmentally friendly, and have a wide range of applications in various fields such as biomedicine [1,2], pharmaceutics [3], energy storage and batteries [4,5], water purification [6], agriculture [7], and other industries. Bottom-up approaches, such as using cellulose nanofibers [8], graphene oxide [9], and bamboo nanofibers [10], have been popular methods for fabricating cellular hydrated materials. However, these approaches often involve complex manufacturing processes, high energy consumption, and the use of toxic chemicals, which can lead to unsatisfactory mechanical properties. Despite the advantages of blending with additional components, a new manufacturing method that is fast, clean, and efficient is needed to produce cellular materials.

To reduce costs and improve efficiency, a top-down approach has been implemented to manufacture cellular materials [11]. Song et al. proposed a method whereby entire blocks of wood are used, chemicals are removed from wood cells through chemical treatment, and freeze drying is employed to shape them into cellular materials. However, traditional chemical treatments that utilize methods such as NaOH-Na_2_SO_3_ and NaClO_2_-H_2_O_2_ treatments tend to destroy the cellulose crystal structure [12]. As a result, the mechanical properties of wood are compromised [13], leading to poor mechanical properties in the final cellular material. Additionally, these methods can produce a lot of pollution, which is not environmentally friendly.

Deep eutectic solvents (DESs) a new class of ionic liquid analogues, which are low-temperature eutectic mixtures composed of hydrogen bond donors (HBD) and hydrogen bond acceptors (HBA), such as quaternary ammonium salts and metal salts, in a certain mole ratio. DESs are green solvents exhibiting low vapor pressure, high thermal stability, low toxicity, and biodegradability [14]. The combined diversity of eutectic solvents allows for adjustment of their physical and chemical properties, providing tremendous potential for industrial applications such as electrochemistry [15], gas absorption [16], extraction [17], and biopharmaceuticals [18]. DESs composed of choline chloride and lactic acid or oxalic acid have been reported to be effective in removing lignin and hemicellulose from wood [16]. Lignin is dissolved in the DES system and subsequently separated, while the ability of C-C chemical bonds in lignin remains unaffected, allowing most of the characteristics and activities of natural lignin to be maintained [19]. Compared to traditional methods, DES delignification treatment does not damage the cellulose structure, and the stiff cell wall becomes more flexible [20]. DES treatment is a safe and efficient process, as it is a physical dissolution process rather than a chemical decomposition process [21], and DESs can be recycled after the termination of reactions. Economically and environmentally, using DES for the delignification of wood enables the production of cellulose-based wood scaffolds, making it a vital process.

In this study, a simple, clean, and scalable top-down approach was employed to fabricate elastic and compressible cellulose wood (also known as elastic material) from natural wood. The process involved treating natural wood with a DES solution to remove a portion of lignin and hemicellulose and soften the cell walls of the wood. An alkali solution was then used to further remove hemicellulose in the wood cell wall. In addition, chemical treatment and freeze drying of the ice templates led to thinner cell walls of the wood that adhered to each other, forming a honeycomb structure. compressible woodis highly porous and can be compressed under high strain (e.g., 70%). However, natural wood is prone to fracture and cannot fully recover to its original state after deformation. In contrast, as shown in Figure 1, compressible wood exhibits a high degree of recoverability, almost fully recovering to its original state even when compressed under high strain (e.g., 70%). To analyze the treatment effect of the reagents, two different methods were employed. The first method involved treating natural wood with DES alone, while the second method involved treating natural wood with both DES and an alkali solution. To distinguish between the two types of wood, the pretreated wood is referred to as NW, and the post-treated wood is called CW. The CW-1 samples were treated with DES alone, while the CW-2 samples were treated with both DES and alkali solution.

## 2. Materials and Methods

### 2.1. Materials

The experiment utilized ash wood (Populus tomentosa Carr.) samples with dimensions of 15 × 15 × 15 mm, which were provided by Yihua Life Technology Co., Ltd. located in Shantou, China. Anhydrous ethanol was sourced from Guangdong Guanghua Sci-Tech Co., Ltd., Guangdong, China, while acetone was provided by Sinopharm Chemical Reagent Co., Ltd., Beijing, China. Choline chloride (AR, 98%) was supplied by Shanghai Canspec Scientific Instruments Co., Ltd., Shanghai, China, and lactic acid (AR, Content 85%~92%) was obtained from Tianjin Fuyu Fine Chemical Co., Ltd., Tianjin, China. Sodium hydroxide (AR, 98%) was procured from Shanghai Aladdin Biochemical Technology Co., Ltd., Shanghai, China. All chemicals used were of analytical grade.

### 2.2. Experimental Procedure for Compressible Wood

#### 2.2.1. Preprocessing

To prepare the ash wood blocks for the subsequent experiments, a preprocessing step was conducted. First, the wood blocks were immersed in a solution of anhydrous ethanol and acetone at a 1:1 volume ratio for 10 h to remove gums and resins in the wood. Next, the blocks were air-dried in an oven at 102 °C for 6 h to completely remove the solvents and any residual moisture. This preprocessing step ensured that the ash wood blocks were clean and free of impurities that could interfere with the subsequent treatments.

#### 2.2.2. DES Solution Treatment

To make deep eutectic solvents, chloride choline and lactic acid were mixed in a 1:5 molar ratio, added to a beaker, and stirred with a magnetic stirrer at 60 °C until a colorless and transparent liquid was formed. Detailed reaction equations are shown in Figure 2. Then, the Ash wood samples were fully immersed in the DES solution (at a mass ratio of 1:10) and heated in a 120 °C oil bath for 5 h, resulting in DES-treated wood. Studies have shown that as the treatment time increases and the temperature rises, more lignin is removed from wood materials . It was discovered that DESs can be synthesized at various molar ratios of HBA to HBD (HBA:HBD). The acid containing a hydroxyl group (such as lactic acid) is capable of forming DESs with an HBA ratio of 1:1, while other HBAs may require a ratio of 1:2 or more [22]. In this study, we used lactic acid as HBDs at a molar ratio 1:5; additional HBDs may lead to the dissolution of more lignin. In this experiment, a molar ratio of 1:5 and a high-temperature, short-duration treatment method was adopted to remove lignin while preserving a certain level of mechanical properties of the wood blocks.

#### 2.2.3. Alkaline Solution Treatment

To further modify the properties of the DES-treated wood, the compressible cellulose wood samples were washed with deionized water and subsequently treated with a 5 wt% sodium hydroxide solution for a 3 h reaction at 80 °C (The mass ratio of solid to liquid was 1:10), resulting in alkaline-treated wood. During this process, more lignin and hemicellulose were removed, further enhancing the flexibility and elasticity of the wood.

#### 2.2.4. Freeze Drying

All the samples, were prefrozen at −15 °C for 6 h, then evacuated at −56 °C for 36 h. It is worth noting that dry compressible cellulose wood can be converted back to compressible wood by absorbing approximately 60% of its weight in water. This was achieved by immersing the dry compressible wood into water. The water not only penetrated the porous cell wall but also filled in empty spaces within the compressible wood.

### 2.3. Characterization

#### 2.3.1. Scanning Electron Microscopy

To observe the morphological characteristics of the sample, an FEI Quanta 200 scanning electron microscope (SEM) was utilized with an acceleration voltage of 30 kV. First, the natural wood and compressible wood samples were cut into small pieces using a glass knife while maintaining the integrity of the sample structure. Next, the pieces were fixed onto conductive carbon glue with tweezers and sprayed with gold films for 30 s using a vacuum coating instrument. Finally, the vessels, cell walls, and cell cavities of the sample were observed with SEM.

#### 2.3.2. X-ray Diffraction and Fourier Infrared Spectroscopy

To characterize the molecular structure of the samples, an X-ray diffractometer (Ultima IV, from Rigaku Co., Ltd., Tokyo, Japan) was utilized. First, all samples were crushed into small, uniform powders, and an appropriate amount was flattened in the sample tank. The X-ray diffractometer was then used to scan the sample with a scanning voltage of 40 kV, a scanning range of 2θ = 5–60°, and a scanning speed of 10°/min. The crystallinity index (*I_c_*) was determined using Formula (1) based on the height of the 200 peak (*I*_200_, 2θ = 22.5°) and the minimum between the 200 and 110 peaks (*I_am_*, 2θ = 18°). It is worth noting that *I*_200_ represents both crystalline and amorphous materials, while Iam only represents amorphous material [23].
(1)$${Ι}_{c}=\left[\frac{{Ι}_{200}-{Ι}_{am}}{{Ι}_{200}}\right]\times 100\%$$

To characterize the infrared spectra of functional groups of samples, a Fourier transform infrared spectrometer (VERTEX 80V, from Bruker Co., Ltd., Bremen, Germany) was used. The method involved grinding samples into wood powder of 80 to 100 mesh, mixing and tableting the powder with KBr, then placing it into a sample pool. The sample was then scanned to obtain the infrared spectrum, allowing for the identification of functional groups present in the samples.

#### 2.3.3. Chemical Composition Content Test

To determine the cellulose, hemicellulose, and lignin contents of the samples, the laboratory analytical procedure (LAP) described by the National Renewable Energy Laboratory was employed. First, the wood powders were hydrolyzed with 72% concentrated sulfuric acid, and the resulting solution was diluted to 4% and filtered using a G3 glass sand funnel. The acid-insoluble lignin content was calculated by weighing the filtered residue, while the acid-soluble lignin content was determined by measuring the ultraviolet absorbance of the filtered filtrate. The sugar content was then analyzed by high-performance liquid chromatography (HPLC) after diluting the filtrate by a certain multiple. Finally, the contents of cellulose, hemicellulose, and lignin were calculated according to Formula (2).
(2)$$\mathrm{content}=\frac{M_{S}}{M_{W}}\times 100\%$$

In the formula, *M_S_* refers to the mass of each chemical component in the sample, and *M_W_* refers to the total mass of the samples.

#### 2.3.4. Mechanical Performance Test

A compression test was conducted using a computer-controlled electronic universal testing machine (AG-IC/100KN, Shimadzu Co., Ltd., Kyoto, Japan). The samples were loaded at a speed of 2 mm/min until the compressive strain reached 70%, and the resulting stress–strain curves were recorded by a computer. Loading–unloading cyclic compression experiments were also performed using the same machine at 30% compression strain and a loading speed of 4 mm/min. The data obtained from these experiments were used to analyze the mechanical properties of the samples.

#### 2.3.5. Laser Confocal Test

The fluorescence emission spectra of lignin were acquired using a confocal laser microscope (LSM710, Carl Zeiss AG Co., Ltd., Kupferzell, Germany) by exciting the lignin with a laser at a wavelength of 488 nm. The samples were cut into slices that were 0.5 mm thick and placed on a slide. A cover slide was then applied, and the sample was observed under the confocal laser microscope to obtain the fluorescence emission spectra of the lignin.

## 3. Results and Discussion

### 3.1. Morphology Analysis

Ash wood is a widely available, fast-growing tree species in China with several advantageous characteristics, including wide adaptability, a long annual growth period, and fast production speed. Due to these features, it was selected as the experimental material for this study. The morphology of samples can be visualized by SEM images. Specifically, NW mainly consists of tough wood fibers, vessels, and ray cells; the tough wood fiber cells have a small cell cavity and a thick cell wall and are randomly distributed among the uniform single-row xylem rays (Figure 3a,b). After DES treatment, the cellular structure was maintained, but cracks appeared between cells, and the cell wall collapsed (Figure 3c,d) [20]. It is worth noting that no significant thinning of cell walls was observed, perhaps because DES selectively removes lignin between cells and does not affect the structural components of cell walls significantly. Upon treatment with NaOH solution, cell walls became thinner, and adjacent cell walls stuck together, forming a highly porous network structure.

### 3.2. Chemical Structure Analysis

As shown in Figure 4a, the characteristic absorption peaks of NW are the same as those reported in previous studies, with absorption peaks at 3327 cm^−1^ (O–H stretching vibration), 2915 cm^−1^ (C–H stretching vibration), 1032 cm^−1^ (ether bond vibration), 1735 cm^−1^ (C=O stretching vibration), 1242 cm^−1^ (C–O stretching vibration), 1505, and 1592 cm^−1^ (aromatic nucleus skeleton vibration). The infrared spectra presented in Figure 4a show that a significant intensity reduction appears in peaks of CW-2 at 1735 cm^−1^, which represents hemicellulose acetyl groups, and at 1238 cm^−1^, which represents the C–O stretching vibration in lignin and hemicellulose, indicating the partial removal of hemicellulose. The characteristic absorption peak strengths of lignin at 1590 and 1505 cm^−1^ were also reduced, indicating the partial removal of lignin. In addition, compared with NW, the absorption peak intensities of CW-1 at 1238 cm^−1^ and 1735 cm^−1^ increased, which may be due to the residual lactic acid in the wood.

The crystal structure of cellulose shows the characteristic diffraction peaks of cellulose in the Figure 4b.Tthe positions of the main diffraction peaks of samples are almost the same at 2θ = 15.6° (101 plane), 22.5° (002 plane), and 34.4° (040 crystal plane), indicating that both CW-1 and CW-2 retain the crystal structure of cellulose I [24]. As shown in Figure 4b, CW-2 has the highest crystallinity (about 80%), followed by CW-1 (about 78%), and NW has the lowest crystallinity of about 62%. The diffraction peak intensity of CW increases at 2θ = 15.6 and 22.5°, which may be due to the fact that after partial lignin removal by DES and alkali solution, the cellulose chain tightly wrapped by lignin remains basically parallel, making it easier to crystallize into cellulose type I, and under high-temperature and acidic environment, the hydroxyl group in DES and the primary hydroxyl group in cellulose participate in the alkylation of cellulose selectively. Additionally, the removal of substances such as lignin, hemicellulose, and amorphous cellulose dominated the pretreatment process, resulting in thinning of the cell wall, which is consistent with the results presented in Figure 3 [25].

### 3.3. Mechanical Performance

The compressive stress–strain curves of samples were obtained as shown in Figure 5. The curves exhibited typical three-state behavior, with a low-strain linear elastic deformation region of less than 4%, followed by a steady-state region in which the stress between 4 and 60% strain remained relatively constant, and finally, a region where the stresses increased sharply with strain. NW showed a significant inflection point at a strain of about 4%, indicating the onset of nonlinear elastic deformation. However, after the stress was released, the strain could not be fully recovered, and partial plastic deformation occurred. In contrast, CW required less stress to achieve the same strain and benefited from its honeycomb structure and hydration network, showing rapid recovery properties when releasing the applied pressure [26]. In contrast, it was almost impossible for NW to regain its shape when released from the same compressive strain. To assess their fatigue resistance, the samples were subjected to 50 load–unload cyclic compression tests at a constant strain of 30% (Figure 5). The results show that after 50 compression cycles, the height retention rate of CW-1 and CW-2 was about 90% and 96%, respectively, indicating good mechanical compressibility and fatigue resistance. Moreover, the plastic deformation produced by CW was much smaller than that of NW (10% and 4%, respectively), demonstrating that CW has superior mechanical properties over NW.

### 3.4. Chemical Composition Content Analysis

Wood cell walls are composed of linear polysaccharide cellulose, heterohemicellulose, and structurally variable lignin, which are interconnected through hydrogen and covalent bonding [27]. The cellular interlocking structure of wood is critical, and chemical treatment and freeze drying can impact this structure. Figure 6 shows that the content of cellulose decreased from 45.4% (NW) to 40.2% (CW-1) to 39.7% (CW-2), while the content of hemicellulose decreased from 13.9% (NW) to 9.3% (CW-1) to 3.1% (CW-2). The content of lignin decreased from 29.6% (NW) to 25.6% (CW-1) to 18.7% (CW-2). The DES treatment dissolved and removed part of the lignin and hemicellulose and a small amount of cellulose. The subsequent alkaline solution treatment further removed part of the lignin and hemicellulose, particularly a large amount of hemicellulose in the cell wall, which led to thinning of the cell wall, as shown in Figure 3.

### 3.5. Confocal Fluorescence Figure

The distribution of lignin in wood is not uniform, with a general trend of lower content at collection sites closer to the top of the plant. Lignin is concentrated in the intercellular layer, followed by the secondary wall inner layer, and the concentration of lignin in the cell interior is the lowest. To determine the content and location of lignin in cells, confocal laser testing is necessary. Confocal laser imaging allows for the estimation of lignin deposition in plant samples and provides spatial quantitative information on the relative amount and composition of lignin [28]. Qualitative observation of the microdistribution of lignin content can be achieved using blue light with a wavelength of 488 nm. The brighter the fluorescence, the higher the density and concentration of lignin. Figure 7a shows that NW has the highest brightness, followed by CW-1 (Figure 7b), and CW-2 is the darkest (Figure 7c). These results are consistent with the chemical composition content determination shown in Figure 6. In NW, lignin is distributed heavily in the cell wall and cell cavity. In CW-1, the confocal fluorescence figure shows a general dimming in brightness, indicating that some lignin was removed. In CW-2, fluorescence intensity is barely observed within the cells, with only a faint glow seen between the cells, indicating that lignin was almost completely removed from the cell wall.

### 3.6. Thermal Property

Figure 8 shows the typical TGA and DTG curves of natural wood and compressible wood. All curves exhibit a minor weight loss below 100 °C, which can be attributed to water evaporation. Above 200 °C, the weight loss rate gradually increased, and a distinct weight loss was observed between 200 and 400 °C. The first weight loss stage was due to water evaporation, while the first degradation event occurred between 200 and 300 °C and can be attributed to the decomposition of hemicelluloses and the slower decomposition of lignin. The second degradation stage above 300 °C can be attributed to the degradation of cellulose. These findings are consistent with previously reported results [29,30].

The Table 1 shows the rmogravimetric data of NW and CW. The T_max_, which refers to the temperature at which the maximum weight loss occurs during decomposition, was found to be lower for the compressible wood compared to the natural wood. In addition, the charcoal residue of CW-2 increased by 42.7% compared to NW. This increase in char residue can be attributed to the removal of low-molecular-weight polysaccharides and some inorganic matter or extractives during the compressible wood treatment process, resulting in a lower decomposition temperature and a higher amount of char formation [31]. These findings suggest that the flame retardancy of treated wood could be improved.

## 4. Conclusions

In this study, we present a top-down approach for producing long-lasting compressible cellulose materials by chemically treating natural wood with DES and alkaline solutions, followed by freeze drying. The chemical treatment process removes some of the lignin and hemicellulose, which not only softens the cell walls but also causes them to adhere to each other and form a honeycomb structure. The resulting resilient wood can be more easily compressed and exhibits a sustainable rebound. DES is an environmentally friendly chemical that can be recycled after use, making this approach a sustainable option. These compressible wood materials have potential applications in fields such as sensors, directional tissue engineering, and water purification.

## Figures and Tables

**Figure 1 polymers-15-01593-f001:**
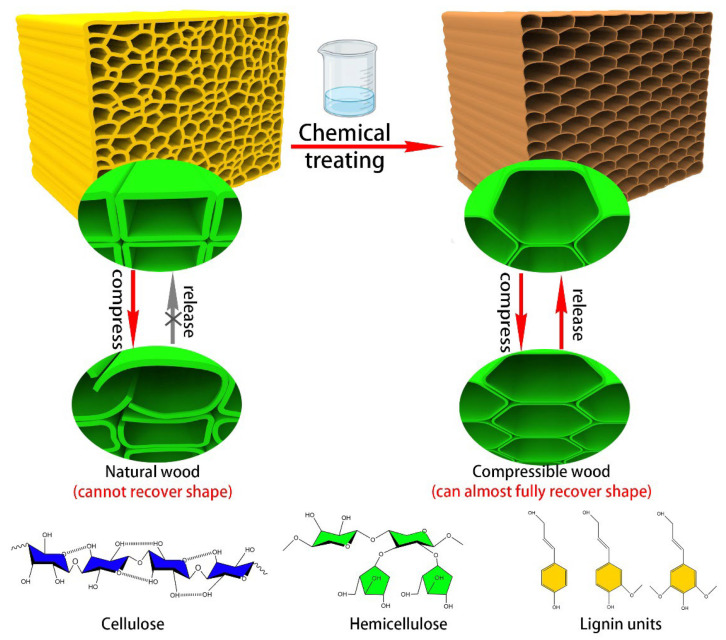
Comparison between natural wood and compressible wood.

**Figure 2 polymers-15-01593-f002:**
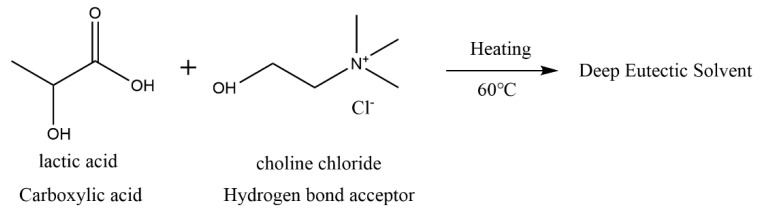
Choline chloride reacted with hydrogen bond acceptor to produce deep eutectic solvents.

**Figure 3 polymers-15-01593-f003:**
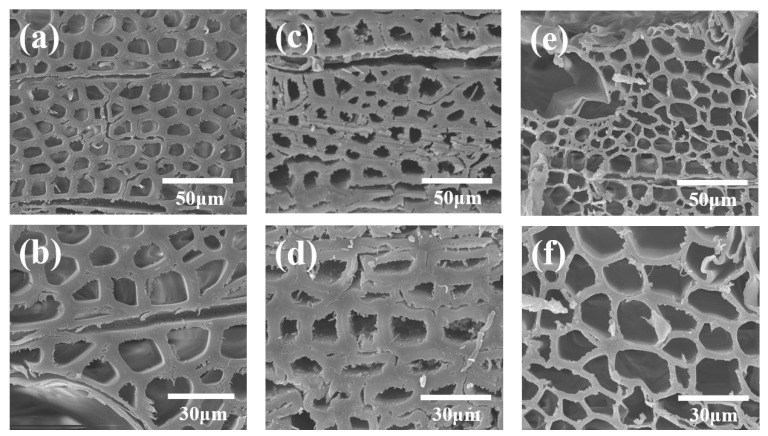
Morphological and structural characterizations of NW and CW. (**a**,**b**) SEM images of NW. (**c**,**d**) SEM images of CW-1. (**e**,**f**) SEM images of CW-2.

**Figure 4 polymers-15-01593-f004:**
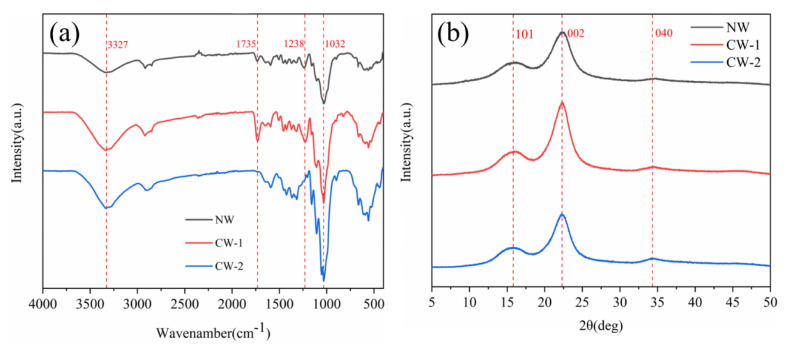
(**a**) FT-IR spectra of NW and CW. (**b**) XRD patterns of NW and CW.

**Figure 5 polymers-15-01593-f005:**
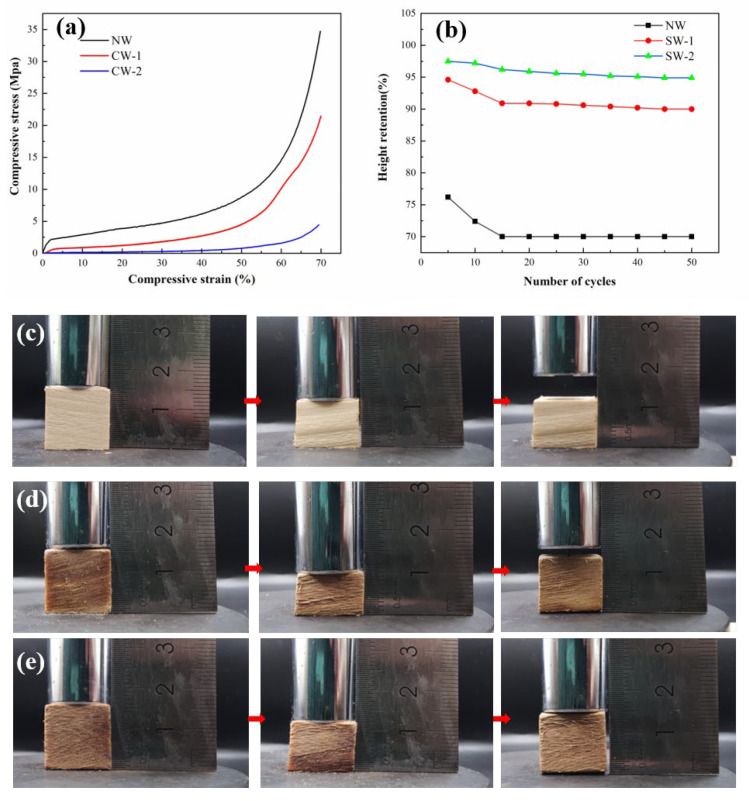
Mechanical properties of NW and CW. (**a**) Stress–strain curves of NW and CW during compression. (**b**) Height retention of NW and CW under cyclic compression with a compression strain of 30%. (**c**) Compression resilience picture of NW. (**d**) Compression resilience picture of CW-1. (**e**) Compression resilience picture of CW-2.

**Figure 6 polymers-15-01593-f006:**
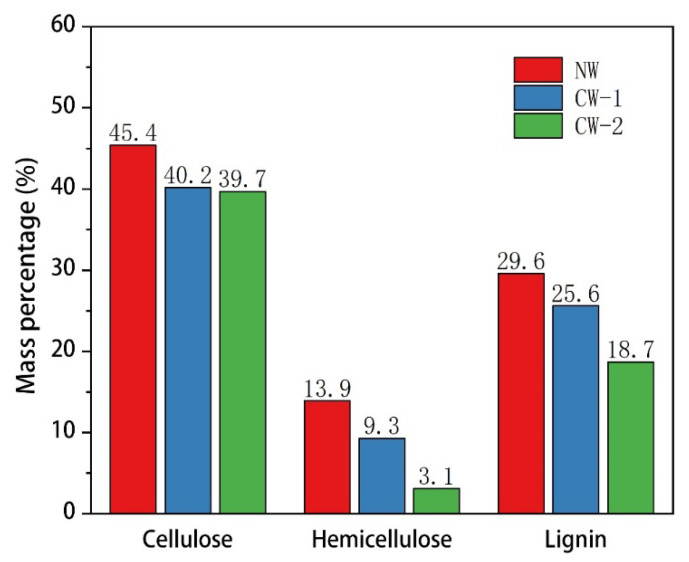
Cellulose, hemicellulose, and lignin contents of NW and CW.

**Figure 7 polymers-15-01593-f007:**
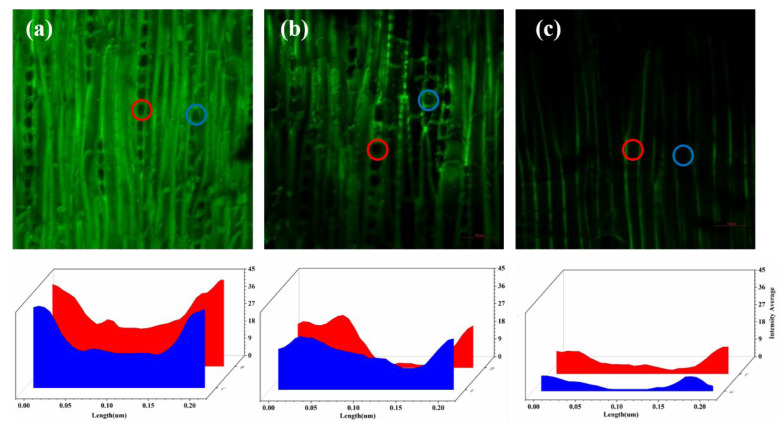
Confocal fluorescence figure of NW and CW under blue light with a wavelength of 488 nm. (**a**) Confocal fluorescence figure of NW. (**b**) Confocal fluorescence figure of CW-1. (**c**) Confocal fluorescence figure of CW-2. Circles indicate the regions of interest for the analysis of the fluorescence intensity by confocal microscopy at corresponding positions of the cells.

**Figure 8 polymers-15-01593-f008:**
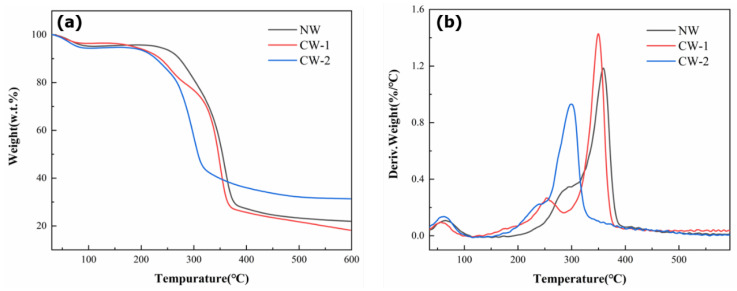
Typical TGA curves (**a**) and DTG curves (**b**) of NW and CW.

**Table 1 polymers-15-01593-t001:** The rmogravimetric data of NW and CW.

Sample	T0 (°C)	Tf (°C)	Tmax (°C)	Residue at 600 °C(%)
NW	249.51	377.38	368.54	21.95
CW-1	212.20	367.24	355.72	18.19
CW-2	207.13	332.01	297.15	31.33

## Data Availability

The data presented in this study are available on request from the corresponding author.

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
