# Peer review of "Compressible Cellulose Wood Prepared with Deep Eutectic Solvents and Its Improved Technology"

_polymers, 2023, doi:10.3390/polym15071593_

Round 1

Reviewer 1 Report

The article Compressible cellulose wood prepared with deep eutectic solvents and its improved technology, aims at the use of deep eutetic solvents in cellulose woord, which generated very interesting properties, and a breakthrough in the area. The article is very interesting and presents excellent results that are well discussed. However, minor adjustments are necessary for it to be accepted.

1. In the chemical composition, the decrease in the composition of lignin, cellulose and hemicellulose is indicated, in Figure 5, after the treatments. However, the authors did not indicate the rest of the composition, which resists the treatment, since there is a considerable decrease in these three indicated components.

2. In TG, authors must substitute weight per mass. And based on the previous information (Item 1), improve the discussion regarding the alteration of the degradation profile.

3. Discuss further why the CW-1 material loses more mass than the NW, since it has less lignin, cellulose and hemicellulose. Perhaps other characterizations will help to better describe these questions.

Author Response

We are so greatful for your kind question

Reviewer 2 Report

This manuscript is about the use of eutectic solvents to prepare compressible woods. I have some suggestions:]

- Please, include the purity of reagents, mainly lactic acid, which normally has 10 - 20% water.

- Why did you use these conditions (5 h, 120.C, 1/5 molar ratio)? Include a reference about it or justify. What was the mass ratio between wood and solvent? Why ChCl/ lactic acid and not other eutectic solvent?

Author Response

We are so grateful for your kind question

Reviewer 3 Report

Before publication, this article needs to resolve the following issues:

- you used many abbreviations that could be avoided; it would be advisable to use only NW, CW-1 and CW-2, so that the text is easy to understand.

  Who is SW?

- Fig 3b shows that CW-1 (red) has the highest crystallinity, followed by CW-2 (blue) and NW (black). they also agree with figure 2.

- As DES is a bifunctional compound (contains quaternary ammonium salt + a reactive OH group) and the treatment is done at a temperature above 100°C, in the presence of an acid (lactic acid), it would be advisable to study the option of alkylation the cellulose, by involving the OH groups from DES and the primary OH from cellulose. The formation of alkyl cellulose could justify the higher crystallinity (fact confirmed by SEM). On an FTIR spectrum as small as in Fig. 3a, it is not clear/visible if CH3 groups from DES are present.

 - In Fig. 3a the peaks at 1735 cm-1 and 1238 cm-1 could come from the lactic acid left in the wood, from the treatment with DES_+ lactic acid.

- In CW-2, larger amorphous areas appear because NaOH has a cleaning effect, determining the removal of hemicellulose, pectin, waxes, etc. The lower crystallinity of the treated wood determines a better relaxation.

Author Response

(The authors gave the same response as above.)

Round 2

Reviewer 2 Report

The authord answered my questions, but in diagram included eutectic is miswritten, make this correction please

Author Response

Thank you gor your kind suggestion

I have change the "eutectiv" in Figure 2 to "eutectic"

Reviewer 3 Report

The article can be accepted for publication because it has been significantly improved.

Author Response

Thank you for your questions and suggestions. All of your perspectives are crucial to this article.